# Incidence and factors associated with reoperation after rotator cuff repair in Korea: A nationwide cohort study

Yee Ran Lyu[1], Eunkyoung Ahn[2], Doori Kim[3], Changsop Yang[1], Mi Hong Yim[4]*

1 Korean Medicine Science Research Division, Korea Institute of Oriental Medicine, Daejeon, Republic of Korea, 2 Korean Medicine Data Division, Korea Institute of Oriental Medicine, Yuseong-gu, Daejeon, Republic of Korea, 3 Jaseng Spine and Joint Research Institute, Jaseng Medical Foundation, Seoul, Republic of Korea, 4 Digital Health Research Division, Korea Institute of Oriental Medicine, Daejeon, Republic of Korea

* mh.yim@kiom.re.kr

## Abstract

### Introduction

With the growing interest in rotator cuff repair (RCR), substantial evidence on factors associated with reoperation is crucial for decision-making. This retrospective longitudinal cohort study aimed to investigate the rate of rotator cuff reoperation after repair in Korea and identify factors associated with reoperation while considering the medical services implemented in Korea using the Korean National Health Insurance Database from 2011 to 2021.

### Methods

Reoperation rates in patients who underwent RCR during a 3.5-year follow-up period were analyzed using Kaplan–Meier analysis. Pre-, intra-, and post-operative factors were analyzed using Cox proportional hazards regression model analyses.

### Results

This study included 221,361 patients, among whom 15,089 (6.82%) underwent rotator cuff reoperation during the follow-up period, reflecting mid- to long-term reoperation (6 months to 3.5 years after surgery), rather than overall surgical failure. Sex, age, health insurance type, healthcare institution type, comorbidities, primary RCR type, arthroscopy use, primary diagnosis of primary RCR, and time to the first outpatient visit for Korean medicine healthcare or conventional medical healthcare were associated with rotator cuff reoperation.

**Data availability statement:** The data used in this study are owned by the Health Insurance Review and Assessment Service (HIRA) of Korea and cannot be publicly shared due to legal and institutional restrictions. Qualified researchers may apply for access to the same HIRA claims database through the HIRA Healthcare Big Data Hub (https://opendata.hira.or.kr) in accordance with HIRA's official procedures. To comply with long-term data accessibility requirements, data access requests related to the minimal dataset generated for this study may be directed to the HIRA Big Data Management Office (email: opendata@hira.or.kr, phone: +82-33-739-5421), which serves as a non-author institutional point of contact. Additional aggregated analytical materials used in this study are available from the corresponding author upon reasonable request.

**Funding:** The authors declare that they have no competing interests. This research was supported by the Korea Institute of Oriental Medicine (Grant No. KSN2122211). The funder had no role in study design, data collection and analysis, decision to publish, or preparation of the manuscript.".

## Conclusion

Our results may help perioperative decision-making and postoperative management strategies related to reoperation risk, while considering patient characteristics, and may support counselling regarding the timing of postoperative care.

## Introduction

Rotator cuff tears are a common shoulder problem, accounting for 30–40% of shoulder pain cases [1]. The overall incidence of rotator cuff tears ranges from 5% to 40% and increases from 25% among individuals in their 60s to 50% in patients over 80 years [2]. Rotator cuff tears substantially affect daily functioning in the growing elderly population and impose considerable socioeconomic burdens on the healthcare system [3].

Depending on the type of injury, the management of rotator cuff injuries varies from non-operative therapy to surgical repair. Conservative therapies are recommended to prevent functional impairment. For patients with complete tendon rupture or in whom conservative management of symptomatic tears has failed, surgical repair is indicated to provide pain relief and functional improvement [4]. Recently, the frequency of rotator cuff tear repair has increased by twofold over 10 years, with rotator cuff tear repair becoming a common procedure worldwide and being affected by several major shifts in the technique, with a large increase in arthroscopic repair over open and mini-pen techniques [5]. The outcomes of surgical repair have also been reported to be mostly positive and satisfactory with pain control and functional improvements [6,7].

Despite the outstanding clinical outcomes of rotator cuff repair (RCR) and the evolving surgical techniques, postoperative adverse outcomes remain a concern. In particular, reoperation following RCR has been reported in a wide range of cases (11% to 94%), representing an important clinical event after surgery [8,9]. It should be noted that reoperation does not necessarily indicate structural failure or tendon re-tear, as not all structural failures lead to surgical intervention, and reoperations may occur for various clinical reasons. To date, factors associated with reoperation after RCR remain unclear. Preoperative characteristics such as patient factors, tear and shoulder morphology, intraoperative repair management, and postoperative rehabilitation strategies may be associated with the likelihood of reoperation [10,11].

Among preoperative factors, patient age [12] and medical comorbidities such as diabetes [13], hypercholesterolemia [14], and smoking [15] have been identified as risk factors for rotator cuff pathology. Other factors related to tear and shoulder morphology such as initial tear size (tear dimensions, tear size area, tear thickness) and fatty degeneration of the supraspinatus have also been reported in relation to structural outcomes following RCR [16,17]. The clinical outcomes, complications, and incidence of rotator cuff reoperation vary depending on the repair technique (open or arthroscopic), although no significant differences have been found between them [12,17]. Postoperative management strategies, including early or delayed motion

rehabilitation protocols and adjunctive physical therapies, may also be associated with differences in reoperation risk [18,19]. In Korea, Korean medical interventions such as acupuncture, electroacupuncture, moxibustion, cupping therapies, Korean medicinal physical therapies, and herbal medicines also used in postoperative rehabilitation [20,21]. As multiple clinical and procedural factors may be associated with the incidence of reoperation, understanding the influence of each factor will help in the decision-making process for RCR.

Therefore, this retrospective longitudinal cohort study aimed (1) to investigate the rate of rotator cuff reoperation after repair in Korea and (2) to identify factors associated with reoperation while considering the medical services being implemented in Korea, including both conventional and Korean medical interventions, using the Korean National Health Insurance Database from 2011 to 2021 in an effort to make it more suitable for Korean medical circumstances. Regarding preoperative factors, patient sex, age, health insurance type, region, healthcare institution type, and comorbidities were analyzed. Data on the type of operation code, arthroscopy use, and primary diagnosis were also collected. As for postoperative factors, rehabilitation therapy, physical therapy, injection therapy, acupuncture, electroacupuncture, moxibustion, cupping therapy, Korean medicinal physical therapy, and herbal medicines were analyzed to investigate factors associated with reoperation. Findings from this study may inform evidence-based surgical decision-making and postoperative management strategies related to reoperation risk after RCR.

## Methods

### Data source and study participants

This nationwide retrospective population-based cohort study used research data derived from the Korean Health Insurance Review and Assessment Service (HIRA; M20220426965). The HIRA is a governmental agency that reviews medical care benefit claims submitted by healthcare service providers and assesses the appropriateness of healthcare services [22]. Healthcare providers submit claims to the HIRA, including details on related healthcare services, and the HIRA collects data during the process of reimbursement decisions based on claims under the National Health Insurance (NHI) [23]. The HIRA research data contain information on patients' general sociodemographic characteristics, prescriptions, diagnoses, procedures, surgeries, examinations, medications, and healthcare providers [23]. Because approximately 98% of the entire population is covered by the NHI for healthcare services in Korea, HIRA research data include population-based health information and represent the Korean population better than any other sample data [23]. HIRA research data are also anonymized and encrypted to protect patient privacy before being released to the public. HIRA research data can be accessed after submitting the application and necessary documentation through the official website (https://opendata.hira.or.kr/home.do) following approval through the review process. Data analysis can be conducted either by connecting to the HIRA server remotely or by visiting the analysis center in person. In the present study, HIRA research data from January 2011 to December 2021 were analyzed. Our study obtained approval for review exemption from the Institutional Review Board of the Korea Institute of Oriental Medicine (IRB no.: I-2204/004-001), and was conducted in accordance with the STROBE (Strengthening the Reporting of Observational Studies in Epidemiology) guideline. The requirement for obtaining informed consent was waived by the IRB because the study involved only secondary analysis of previously collected, de-identified national health insurance claims data, posed no additional risk to participants, and all data were fully anonymized prior to access and analysis to protect participant confidentiality. Data were accessed under approval from the Health Insurance Review & Assessment Service between February 13, 2023, and February 12, 2025, and the authors did not have access to any information that could identify individual participants during or after data collection.

Patients who underwent RCR from January 2014 to June 2018 were included in this study. RCR patients were defined as those who underwent procedures coded N0935 (acromioplasty), N0936 (acromioplasty and repair of ruptured shoulder rotator cuff, primary repair), N0937 (acromioplasty and repair of ruptured shoulder rotator cuff, myoplasty, and tendoplasty), or N0938 (acromioplasty and repair of ruptured shoulder rotator cuff, complex). All patients underwent a 3-year washout period before RCR and a 3.5-year follow-up period after RCR. Primary RCR was defined as the first RCR

procedure performed in patients with no history of RCR during the three years preceding the index procedure. During the 3 years preceding the primary RCR, all patients were observed to determine whether they had undergone any previous RCR. From postoperative day 1–6 months after the primary RCR, all patients were monitored to determine whether they had received healthcare services with a primary diagnosis of a disease code that may be used in the treatment of shoulder and rotator cuff disease or injury associated with rotator cuff surgery (Supplementary Table 1), as described in previous studies [20,24]. From 6 months to 3.5 years after primary RCR, all patients were monitored to verify whether they had undergone rotator cuff reoperation.

A total of 245,951 patients who underwent RCR from January 2014 to June 2018 were analyzed in this study. Patients aged <19 years, those who underwent reoperation within 6 months after acromioplasty or RCR, and those who underwent prior acromioplasty or RCR within 3 years before the index surgery were excluded. The final cohort comprised 221,361 patients (206,272 without reoperation and 15,089 with reoperation; Fig 1).

## Outcome and other variables

The outcome variable of interest was whether the patients underwent reoperation during the follow-up period after primary RCR. Patients who underwent rotator cuff reoperation were defined as those who underwent subsequent procedures coded as N0935, N0936, N0937, or N0938, identical to the coding for primary RCR. Reoperation was defined as any subsequent surgical intervention performed after the index RCR, regardless of the underlying clinical indication. Patients

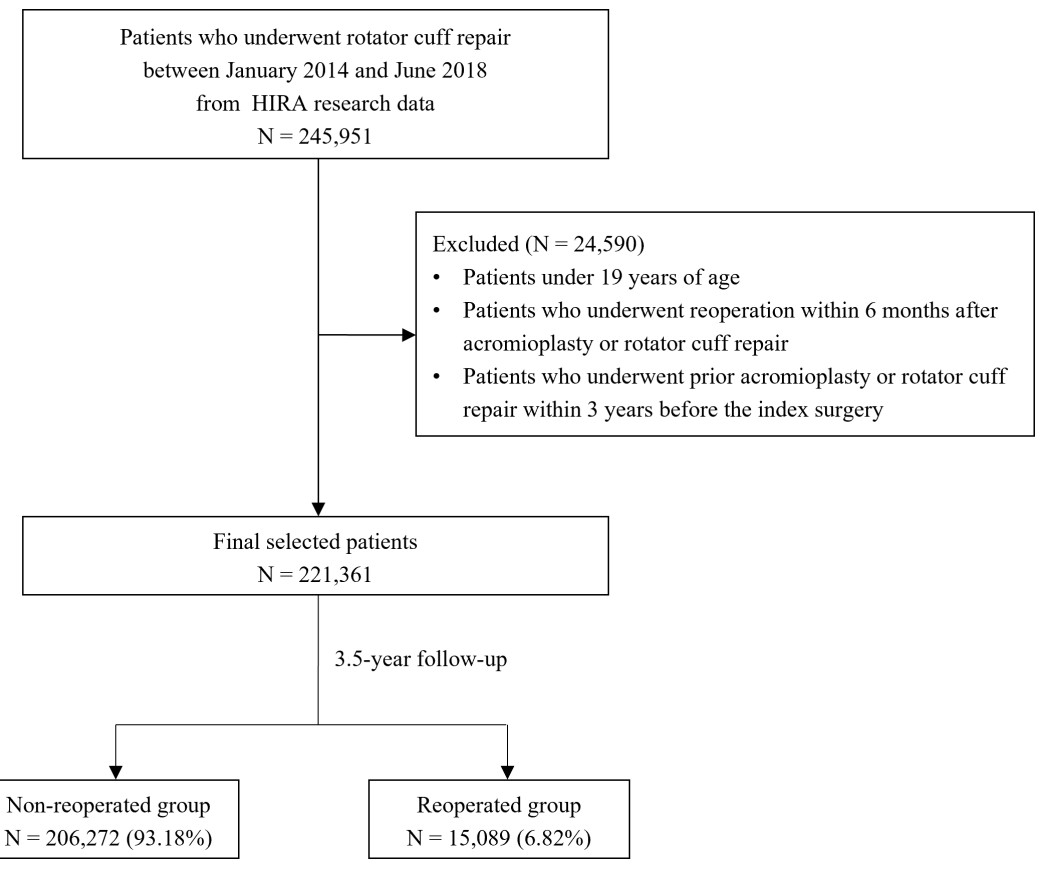

**Fig 1. Flow chart of the study sample selection process.**

who underwent reoperation were identified as those who received subsequent rotator cuff procedures coded as N0935, N0936, N0937, or N0938 after the primary RCR. These procedural codes are standardized within the Korean National Health Insurance system and have been used in nationwide claims-based studies to identify rotator cuff surgeries, supporting their validity for use in administrative data research [25].

Potential factors associated with reoperation were determined based on preoperative, intraoperative, and postoperative factors. Preoperative factors pertained to patients' preoperative characteristics at baseline at the time of primary RCR, except for comorbidities. Comorbidities were evaluated during the 1-year period preceding the primary RCR. Preoperative factors included sex (men or women), age (19–44, 45–54, 55–64, or 65 years or older), region (Seoul/Gyeonggi/Incheon, Gangwon, Daejeon/Chungcheong/Sejong, Busan/Daegu/Ulsan/Gyeongsang, or Gwangju/Jeolla/Jeju), health insurance type (employee/local or medical aid/others), healthcare institution type (tertiary hospital, general hospital, hospital, or clinic), and comorbidities. Comorbidities were assessed using the Charlson Comorbidity Index (CCI) [26], calculated based on 17 diseases converted to ICD-10 codes and their corresponding weights [27]. Intraoperative factors encompassed variables related to the primary RCR, collected at baseline at the time of the primary RCR. Intraoperative factors included the type of surgery (N0935, N0936, N0937, or N0938), arthroscopy use, and primary diagnosis at the time of surgery. Arthroscopy use was determined by categorizing patients with and without the examination code E7500. The primary diagnosis at the time of surgery was classified as ICD-10 code M75 (shoulder lesions), S46 (injury of muscle, fascia, and tendon at shoulder and upper arm levels), or others. Postoperative factors included healthcare utilization with a primary diagnosis of shoulder and rotator cuff disease assessed within 6 months after the primary RCR. Due to limitations in the claims data, postoperative factors included only reimbursable procedures, with those with low utilization frequency excluded from the analysis. For Korean medicine healthcare (KMHC) use, we evaluated whether the patients received six or more sessions of acupuncture, electroacupuncture, moxibustion, cupping therapy, or Korean medicinal physical therapy, considering that the minimum number of treatments in previous studies is 6 times [28]. The use of herbal medicines was determined based on whether the medicines were prescribed at least once. The time to first outpatient visit for KMHC use after primary RCR was categorized as within 6 weeks, 6–12 weeks, 12 weeks to 6 months, or no visit, according to the general classification of postoperative treatment, for which the goals and regimens of treatment are differently applicated [29]. Similarly, for conventional medical healthcare (CMHC), we assessed whether patients received six or more sessions of rehabilitation, physical, or injection therapy. The time to the first outpatient visit for CMHC after primary RCR was categorized using the same criteria as for Korean medicine.

## Statistical analysis

Data were analyzed using a remote connection to the HIRA server. All statistical analyses were performed using SAS Enterprise Guide version 9.4.2 (SAS Institute Inc., Cary, NC, USA) for data preprocessing and R Server version 3.5.1 (R Foundation for Statistical Computing, Vienna, Austria) for statistical modeling and inference. Statistical significance was determined using two-tailed tests, with a significance level of 0.05.

General characteristics of the reoperated and non-reoperated groups were compared using Pearson's chi-squared test because all variables were categorical. Results were presented as frequencies and column proportions. The incidence rates of rotator cuff reoperation were calculated semiannually for each category of potential risk factors using Kaplan–Meier analysis and compared using log-rank tests. The results were reported as semiannual cumulative incidence rates of rotator cuff reoperation. Risk factors associated with rotator cuff reoperation were identified using Cox proportional hazards regression model analyses [30]. Unadjusted analysis was performed using a univariate Cox proportional hazards regression model to evaluate the association between each individual risk factor and reoperation. To examine the association between multiple risk factors and reoperation, an adjusted analysis was conducted using a multivariate Cox proportional hazards regression model. Candidate risk factors for the multivariate model included most of preoperative, intraoperative, and postoperative factors described above, namely sex, age group, health insurance type, healthcare

institution type, CCI score, type of surgery (N0935, N0936, N0937, or N0938), arthroscopy use, primary diagnosis, use of KMHC (including acupuncture, electroacupuncture, herbal medicine, rehabilitation therapy, and Korean medicinal physical therapy), timing to the first outpatient visit for KMHC, and timing to the first outpatient visit for CMHC. The final model was determined using a stepwise variable selection procedure. In addition to the main multivariate Cox proportional hazards model, we conducted a supplementary sensitivity analysis in which the original four CCI categories (0, 1, 2, and ≥3) were reclassified into a dichotomous variable (0 vs. ≥ 1) to assess the stability of the estimated associations. The proportional hazards assumption was evaluated using Schoenfeld residuals and confirmed by visual inspection of log–log plots. Multi-collinearity in the adjusted model was assessed using the generalized variance inflation factor (GVIF), which is applicable to categorical variables with multiple degrees of freedom (DF) [31]. The scaled GVIF, adjusted for DF by taking the GVIF to a power of 1/(2DF), was employed to enable a direct comparison among variables [31]. The results were expressed as hazard ratios (HRs) with corresponding 95% confidence intervals (CIs).

## Results

### General characteristics between the reoperated and non-reoperated groups

A total of 221,361 patients were included in this study, among whom 15,089 (6.82%) underwent rotator cuff reoperation during the follow-up period (i.e., from 6 months to 3.5 years after the primary RCR). The reoperated and non-reoperated groups significantly differed in terms of sex, age, region, health insurance type, healthcare institution type, comorbidities, primary RCR type, arthroscopy use, primary diagnosis of primary RCR, receipt of physical therapy, receipt of injection therapy, time to the first outpatient visit for KMHC, and time to the first outpatient visit for CMHC. At baseline, higher proportions were observed in the reoperated group than in the non-reoperated group with respect to the following characteristics: male sex (52.05% for reoperated vs. 50% for non-reoperated); 45–54-year age group (31.8% vs. 29.22%) and 55–64-year age group (39.72% vs. 34.65%); Daejeon/Chungcheong/Sejong region (7.39% vs. 6.98%) and Gwangju/Jeolla/Jeju region (16.56% vs. 14.61%); health insurance type of medical aid/others (3.88% vs. 2.89%); healthcare institution types of general hospital (21.43% vs. 20.39%) and clinic (6.32% vs. 5.79%); and CCI scores of 1 (30.84% vs. 29.92%), 2 (15.69% vs. 13.73%), and 3 or more (12.13% vs. 11.1%). The proportions of the primary RCR types N0937 (16.3% vs. 13.71%) and N0938 (5.24% vs. 3.83%) were higher in the reoperated group than in the non-reoperated group. At the time of primary RCR, the proportions of individuals undergoing arthroscopy (0.44% vs. 0.31%), those with a primary diagnosis of M75 (70.23% vs. 69.53%), or those with a primary diagnosis of S46 (23.36% vs. 22.52%) were greater in the reoperated group than in the non-reoperated group. The proportion of patients who received physical therapy (42.24% vs. 40.93%) or injection therapy (1.02% vs. 0.59%) six times or more and those who made their first outpatient visit for KMHC (5.53% vs. 4.48%) or CMHC (2.32% vs. 1.63%) between 12 weeks and 6 months after primary RCR were higher in the reoperated group than in the non-reoperated group (Table 1).

### Incidence of reoperation after RCR

The overall cumulative incidence of rotator cuff reoperation was 6.82% over the follow-up period, ranging from 6 months to 3.5 years after primary RCR. There were significant differences in the incidence of rotator cuff reoperation according to the individual risk factors of sex, age, health insurance type, healthcare institution type, comorbidities, primary RCR type, arthroscopy use, primary diagnosis of primary RCR, receipt of physical therapy, time to first outpatient visit for KMHC, and time to first outpatient visit for CMHC. High rotator cuff reoperation rates were found in the following characteristics: men (3.5-year cumulative incidence rate of reoperation, 7.08%); age of 55–64 years (7.74%); health insurance type of medical aid/others (8.95%); healthcare institution types of clinic (7.39); CCI scores of 2 (7.71%); the primary RCR type of N0938 (9.09%); arthroscopy (9.44%); primary diagnosis of S46 (7.05%); receipt of physical therapy (7.02%); and first outpatient visit for KMHC (8.27%) or CMHC (9.41%) between 12 weeks and 6 months after the primary RCR (Table 2).

**Table 1. General characteristics between the reoperated and non-reoperated groups.**

| Variables | Total | Non-reoperated | Reoperated | P value |
|---|---|---|---|---|
| N | 221361 | 206272 | 15089 | |
| Sex | | | | |
| Men | 110994 (50.14) | 103140 (50) | 7854 (52.05) | <.001 |
| Women | 110367 (49.86) | 103132 (50) | 7235 (47.95) | |
| Age | | | | |
| 19-44 | 32826 (14.83) | 31062 (15.06) | 1764 (11.69) | <.001 |
| 45-54 | 65077 (29.4) | 60279 (29.22) | 4798 (31.8) | |
| 55-64 | 77467 (35) | 71473 (34.65) | 5994 (39.72) | |
| 65 or older | 45991 (20.78) | 43458 (21.07) | 2533 (16.79) | |
| Region | | | | |
| Seoul/Gyeonggi/Incheon | 116459 (52.61) | 108796 (52.74) | 7663 (50.79) | <.001 |
| Gangwon | 2761 (1.25) | 2584 (1.25) | 177 (1.17) | |
| Daejeon/Chungcheong/Sejong | 15509 (7.01) | 14394 (6.98) | 1115 (7.39) | |
| Busan/Daegu/Ulsan/Gyeongsang | 54003 (24.4) | 50367 (24.42) | 3636 (24.1) | |
| Gwangju/Jeolla/Jeju | 32629 (14.74) | 30131 (14.61) | 2498 (16.56) | |
| Health insurance type | | | | |
| Employee/local | 214814 (97.04) | 200311 (97.11) | 14503 (96.12) | <.001 |
| Medical aid/others | 6547 (2.96) | 5961 (2.89) | 586 (3.88) | |
| Healthcare institution type | | | | |
| Tertiary hospital | 18512 (8.36) | 17357 (8.41) | 1155 (7.65) | <.001 |
| General hospital | 45288 (20.46) | 42055 (20.39) | 3233 (21.43) | |
| Hospital | 144660 (65.35) | 134913 (65.41) | 9747 (64.6) | |
| Clinic | 12901 (5.83) | 11947 (5.79) | 954 (6.32) | |
| CCI score | | | | |
| 0 | 99574 (44.98) | 93336 (45.25) | 6238 (41.34) | <.001 |
| 1 | 66376 (29.99) | 61723 (29.92) | 4653 (30.84) | |
| 2 | 30694 (13.87) | 28327 (13.73) | 2367 (15.69) | |
| 3 or more | 24717 (11.17) | 22886 (11.1) | 1831 (12.13) | |
| N0935 | | | | |
| No | 151225 (68.32) | 140385 (68.06) | 10840 (71.84) | <.001 |
| Yes | 70136 (31.68) | 65887 (31.94) | 4249 (28.16) | |
| N0936 | | | | |
| No | 109085 (49.28) | 101613 (49.26) | 7472 (49.52) | .546 |
| Yes | 112276 (50.72) | 104659 (50.74) | 7617 (50.48) | |
| N0937 | | | | |
| No | 190614 (86.11) | 177985 (86.29) | 12629 (83.7) | <.001 |
| Yes | 30747 (13.89) | 28287 (13.71) | 2460 (16.3) | |
| N0938 | | | | |
| No | 212673 (96.08) | 198374 (96.17) | 14299 (94.76) | <.001 |
| Yes | 8688 (3.92) | 7898 (3.83) | 790 (5.24) | |
| Arthroscopy | | | | |
| No | 220651 (99.68) | 205629 (99.69) | 15022 (99.56) | .007 |
| Yes | 710 (0.32) | 643 (0.31) | 67 (0.44) | |
| Primary diagnosis | | | | |
| Others | 17370 (7.85) | 16403 (7.95) | 967 (6.41) | <.001 |
| M75 | 154015 (69.58) | 143418 (69.53) | 10597 (70.23) | |

*(Continued)*

PLOS One | https://doi.org/10.1371/journal.pone.0350201   May 26, 2026    7 / 19

**Table 1.** (Continued)

| Variables | Total | Non-reoperated | Reoperated | P value |
|---|---|---|---|---|
| S46 | 49976 (22.58) | 46451 (22.52) | 3525 (23.36) | |
| Acupuncture | | | | |
| 5 or less | 214398 (96.85) | 199790 (96.86) | 14608 (96.81) | .777 |
| 6 or more | 6963 (3.15) | 6482 (3.14) | 481 (3.19) | |
| Electro-acupuncture | | | | |
| 5 or less | 219431 (99.13) | 204491 (99.14) | 14940 (99.01) | .124 |
| 6 or more | 1930 (0.87) | 1781 (0.86) | 149 (0.99) | |
| Moxibustion | | | | |
| 5 or less | 219908 (99.34) | 204934 (99.35) | 14974 (99.24) | .106 |
| 6 or more | 1453 (0.66) | 1338 (0.65) | 115 (0.76) | |
| Cupping therapy | | | | |
| 5 or less | 217685 (98.34) | 202862 (98.35) | 14823 (98.24) | .325 |
| 6 or more | 3676 (1.66) | 3410 (1.65) | 266 (1.76) | |
| Korean medicinal physical therapy | | | | |
| 5 or less | 218048 (98.5) | 203195 (98.51) | 14853 (98.44) | .502 |
| 6 or more | 3313 (1.5) | 3077 (1.49) | 236 (1.56) | |
| Herbal medicine | | | | |
| No | 217280 (98.16) | 202443 (98.14) | 14837 (98.33) | .107 |
| Yes | 4081 (1.84) | 3829 (1.86) | 252 (1.67) | |
| Rehabilitation therapy | | | | |
| 5 or less | 153001 (69.12) | 142651 (69.16) | 10350 (68.59) | .151 |
| 6 or more | 68360 (30.88) | 63621 (30.84) | 4739 (31.41) | |
| Physical therapy | | | | |
| 5 or less | 130565 (58.98) | 121849 (59.07) | 8716 (57.76) | .002 |
| 6 or more | 90796 (41.02) | 84423 (40.93) | 6373 (42.24) | |
| Injection therapy | | | | |
| 5 or less | 219985 (99.38) | 205050 (99.41) | 14935 (98.98) | <.001 |
| 6 or more | 1376 (0.62) | 1222 (0.59) | 154 (1.02) | |
| First visit of KMHC | | | | |
| Within 6 weeks | 5341 (2.41) | 4980 (2.41) | 361 (2.39) | <.001 |
| 6-12 weeks | 8091 (3.66) | 7546 (3.66) | 545 (3.61) | |
| 12 weeks to 6 months | 10084 (4.56) | 9250 (4.48) | 834 (5.53) | |
| No visit | 197845 (89.38) | 184496 (89.44) | 13349 (88.47) | |
| First visit of CMHC | | | | |
| Within 6 weeks | 207281 (93.64) | 193401 (93.76) | 13880 (91.99) | <.001 |
| 6-12 weeks | 10218 (4.62) | 9370 (4.54) | 848 (5.62) | |
| 12 weeks to 6 months | 3720 (1.68) | 3370 (1.63) | 350 (2.32) | |
| No visit | 142 (0.06) | 131 (0.06) | 11 (0.07) | |

Abbreviations: CCI, Charlson Comorbidity Index; N0935, acromioplasty; N0936, acromioplasty and repair of ruptured shoulder rotator cuff, primary repair; N0937, acromioplasty and repair of ruptured shoulder rotator cuff, with myoplasty and tendoplasty; N0938, acromioplasty and repair of ruptured shoulder rotator cuff, complex; M75, shoulder lesions; S46, injury of muscle, fascia and tendon at shoulder and upper arm level; KMHC, Korean medicine healthcare; CMHC, conventional medicine healthcare.

Statistical significance was assessed using Pearson's chi-squared tests for categorical variables. Results were presented as frequencies and column proportions.

**Table 2. Semiannual cumulative incidence rate of reoperation after rotator cuff repair.**

| Variables | Annual cumulative incidence rate (%) (95% CI) | | | | | | P value |
|---|---|---|---|---|---|---|---|
| | 1-Year | | 1.5-Year | 2-Year | 2.5-Year | 3-Year | 3.5-Year |
| Total | 1.57 (1.52-1.62) | 2.81 (2.74-2.88) | 3.95 (3.87-4.03) | 4.98 (4.89-5.07) | 5.94 (5.84-6.04) | 6.82 (6.71-6.92) | |
| Sex | | | | | | | |
| Men | 1.59 (1.52-1.66) | 2.87 (2.77-2.97) | 4.05 (3.93-4.16) | 5.15 (5.02-5.28) | 6.16 (6.02-6.3) | 7.08 (6.93-7.23) | <.001 |
| Women | 1.56 (1.48-1.63) | 2.74 (2.64-2.84) | 3.86 (3.74-3.97) | 4.82 (4.69-4.94) | 5.72 (5.58-5.86) | 6.56 (6.41-6.7) | |
| Age | | | | | | | |
| 19-44 | 1.29 (1.17-1.42) | 2.29 (2.13-2.45) | 3.28 (3.09-3.48) | 4 (3.79-4.21) | 4.77 (4.54-5) | 5.37 (5.13-5.62) | <.001 |
| 45-54 | 1.63 (1.53-1.73) | 3.01 (2.88-3.14) | 4.3 (4.15-4.46) | 5.42 (5.25-5.6) | 6.46 (6.27-6.65) | 7.37 (7.17-7.57) | |
| 55-64 | 1.8 (1.7-1.89) | 3.14 (3.02-3.26) | 4.39 (4.24-4.53) | 5.59 (5.42-5.75) | 6.68 (6.5-6.85) | 7.74 (7.55-7.93) | |
| 65 or older | 1.31 (1.21-1.42) | 2.32 (2.18-2.46) | 3.21 (3.05-3.37) | 4.04 (3.86-4.22) | 4.8 (4.6-4.99) | 5.51 (5.3-5.72) | |
| Health insurance type | | | | | | | |
| Employee/local | 1.55 (1.5-1.6) | 2.76 (2.69-2.83) | 3.9 (3.82-3.98) | 4.93 (4.83-5.02) | 5.87 (5.77-5.97) | 6.75 (6.65-6.86) | <.001 |
| Medical aid/others | 2.26 (1.9-2.62) | 4.28 (3.79-4.77) | 5.73 (5.16-6.29) | 6.86 (6.24-7.47) | 8.13 (7.46-8.79) | 8.95 (8.26-9.64) | |
| Healthcare institution type | | | | | | | |
| Tertiary hospital | 1.36 (1.19-1.53) | 2.44 (2.22-2.66) | 3.42 (3.16-3.68) | 4.42 (4.12-4.71) | 5.29 (4.97-5.61) | 6.24 (5.89-6.59) | <.001 |
| General hospital | 1.58 (1.47-1.7) | 2.91 (2.76-3.07) | 4.09 (3.9-4.27) | 5.11 (4.9-5.31) | 6.2 (5.97-6.42) | 7.14 (6.9-7.38) | |
| Hospital | 1.58 (1.52-1.65) | 2.81 (2.72-2.89) | 3.96 (3.86-4.06) | 4.98 (4.87-5.09) | 5.89 (5.77-6.01) | 6.74 (6.61-6.87) | |
| Clinic | 1.74 (1.52-1.97) | 2.96 (2.67-3.25) | 4.17 (3.82-4.51) | 5.4 (5.01-5.79) | 6.57 (6.14-6.99) | 7.39 (6.94-7.85) | |
| CCI score | | | | | | | |
| 0 | 1.42 (1.34-1.49) | 2.54 (2.44-2.64) | 3.6 (3.48-3.71) | 4.54 (4.41-4.67) | 5.45 (5.3-5.59) | 6.26 (6.11-6.42) | <.001 |
| 1 | 1.56 (1.47-1.66) | 2.84 (2.71-2.97) | 4.03 (3.88-4.18) | 5.11 (4.94-5.28) | 6.11 (5.93-6.3) | 7.01 (6.82-7.2) | |
| 2 | 1.81 (1.66-1.96) | 3.23 (3.03-3.43) | 4.56 (4.33-4.79) | 5.74 (5.48-6) | 6.74 (6.46-7.02) | 7.71 (7.41-8.01) | |
| 3 or more | 1.93 (1.76-2.11) | 3.26 (3.04-3.48) | 4.43 (4.17-4.68) | 5.47 (5.19-5.76) | 6.48 (6.17-6.78) | 7.41 (7.08-7.73) | |
| N0935 | | | | | | | |
| No | 1.67 (1.61-1.74) | 2.95 (2.86-3.03) | 4.12 (4.01-4.22) | 5.18 (5.07-5.29) | 6.21 (6.09-6.33) | 7.17 (7.04-7.3) | <.001 |
| Yes | 1.36 (1.27-1.44) | 2.5 (2.38-2.61) | 3.6 (3.47-3.74) | 4.56 (4.41-4.72) | 5.35 (5.19-5.52) | 6.06 (5.88-6.23) | |
| N0936 | | | | | | | |
| No | 1.57 (1.49-1.64) | 2.82 (2.72-2.92) | 3.98 (3.87-4.1) | 5.06 (4.93-5.19) | 5.97 (5.83-6.11) | 6.85 (6.7-7) | 0.538 |

*(Continued)*

Table 2. (Continued)

| Variables | Annual cumulative incidence rate (%) (95% CI) | | | | | | P value |
|---|---|---|---|---|---|---|---|
| | 1-Year | | 1.5-Year | 2-Year | 2.5-Year | 3-Year | 3.5-Year |
| Yes | 1.58 (1.5-1.65) | 2.79 (2.69-2.89) | 3.93 (3.81-4.04) | 4.91 (4.78-5.03) | 5.91 (5.77-6.05) | 6.78 (6.64-6.93) | |
| N0937 | | | | | | | |
| No | 1.52 (1.47-1.58) | 2.73 (2.66-2.8) | 3.86 (3.78-3.95) | 4.86 (4.76-4.96) | 5.8 (5.69-5.9) | 6.63 (6.51-6.74) | <.001 |
| Yes | 1.89 (1.74-2.05) | 3.29 (3.09-3.49) | 4.5 (4.27-4.74) | 5.74 (5.48-6) | 6.84 (6.55-7.12) | 8 (7.7-8.3) | |
| N0938 | | | | | | | |
| No | 1.55 (1.5-1.6) | 2.77 (2.7-2.84) | 3.9 (3.82-3.99) | 4.92 (4.82-5.01) | 5.86 (5.76-5.96) | 6.72 (6.62-6.83) | <.001 |
| Yes | 2.11 (1.8-2.41) | 3.78 (3.37-4.18) | 5.15 (4.68-5.61) | 6.58 (6.06-7.1) | 7.79 (7.23-8.35) | 9.09 (8.49-9.7) | |
| Arthroscopy | | | | | | | |
| No | 1.57 (1.52-1.62) | 2.8 (2.73-2.87) | 3.95 (3.87-4.03) | 4.97 (4.88-5.07) | 5.93 (5.83-6.03) | 6.81 (6.7-6.91) | 0.005 |
| Yes | 2.39 (1.26-3.51) | 4.79 (3.21-6.35) | 5.92 (4.16-7.63) | 7.46 (5.51-9.38) | 8.17 (6.13-10.16) | 9.44 (7.26-11.56) | |
| Primary diagnosis | | | | | | | |
| Others | 1.18 (1.02-1.34) | 2.19 (1.98-2.41) | 3.2 (2.94-3.46) | 4.12 (3.83-4.42) | 4.94 (4.62-5.26) | 5.57 (5.23-5.91) | <.001 |
| M75 | 1.6 (1.54-1.66) | 2.85 (2.77-2.93) | 3.99 (3.89-4.09) | 5.04 (4.93-5.15) | 5.99 (5.87-6.11) | 6.88 (6.75-7.01) | |
| S46 | 1.62 (1.51-1.74) | 2.89 (2.74-3.03) | 4.11 (3.93-4.28) | 5.11 (4.92-5.31) | 6.14 (5.93-6.36) | 7.05 (6.83-7.28) | |
| Acupuncture | | | | | | | |
| 5 or less | 1.57 (1.51-1.62) | 2.8 (2.73-2.87) | 3.95 (3.86-4.03) | 4.98 (4.88-5.07) | 5.94 (5.84-6.04) | 6.81 (6.71-6.92) | 0.73 |
| 6 or more | 1.8 (1.48-2.11) | 3.06 (2.65-3.46) | 4.19 (3.72-4.66) | 5.2 (4.68-5.72) | 6.07 (5.51-6.63) | 6.91 (6.31-7.5) | |
| Electro-acupuncture | | | | | | | |
| 5 or less | 1.57 (1.52-1.62) | 2.8 (2.73-2.87) | 3.95 (3.86-4.03) | 4.97 (4.88-5.06) | 5.93 (5.83-6.03) | 6.81 (6.7-6.91) | 0.104 |
| 6 or more | 2.12 (1.48-2.77) | 3.63 (2.79-4.46) | 4.77 (3.81-5.71) | 6.11 (5.04-7.18) | 6.94 (5.8-8.07) | 7.72 (6.52-8.9) | |
| Herbal medicine | | | | | | | |
| No | 1.57 (1.52-1.63) | 2.81 (2.74-2.88) | 3.95 (3.87-4.04) | 4.99 (4.89-5.08) | 5.95 (5.85-6.05) | 6.83 (6.72-6.93) | 0.107 |
| Yes | 1.49 (1.12-1.87) | 2.77 (2.26-3.27) | 3.87 (3.28-4.46) | 4.8 (4.14-5.46) | 5.37 (4.67-6.06) | 6.17 (5.43-6.91) | |
| Rehabilitation therapy | | | | | | | |
| 5 or less | 1.57 (1.5-1.63) | 2.8 (2.72-2.88) | 3.94 (3.84-4.03) | 4.97 (4.86-5.07) | 5.91 (5.79-6.03) | 6.76 (6.64-6.89) | 0.154 |
| 6 or more | 1.59 (1.49-1.68) | 2.82 (2.69-2.94) | 3.99 (3.84-4.14) | 5.02 (4.86-5.18) | 6.01 (5.83-6.19) | 6.93 (6.74-7.12) | |
| Physical therapy | | | | | | | |
| 5 or less | 1.53 (1.47-1.6) | 2.76 (2.67-2.85) | 3.87 (3.77-3.97) | 4.9 (4.79-5.02) | 5.84 (5.72-5.97) | 6.68 (6.54-6.81) | 0.002 |

*(Continued)*

**Table 2.** (Continued)

| Variables | Annual cumulative incidence rate (%) (95% CI) | | | | | | P value |
|---|---|---|---|---|---|---|---|
| | 1-Year | | 1.5-Year | 2-Year | 2.5-Year | 3-Year | 3.5-Year |
| 6 or more | 1.63 (1.55-1.71) | 2.87 (2.76-2.98) | 4.07 (3.94-4.2) | 5.1 (4.95-5.24) | 6.08 (5.92-6.23) | 7.02 (6.85-7.19) | |
| First visit of KMHC | | | | | | | |
| Within 6 weeks | 1.78 (1.42-2.13) | 2.96 (2.5-3.41) | 4.1 (3.57-4.63) | 5.09 (4.5-5.68) | 5.97 (5.33-6.61) | 6.76 (6.08-7.43) | <.001 |
| 6-12 weeks | 1.68 (1.4-1.96) | 2.68 (2.33-3.03) | 3.99 (3.56-4.42) | 5.15 (4.67-5.63) | 6.06 (5.53-6.57) | 6.74 (6.19-7.28) | |
| 12 weeks to 6 months | 2.14 (1.86-2.42) | 3.7 (3.33-4.07) | 5.05 (4.62-5.47) | 6.12 (5.65-6.59) | 7.28 (6.77-7.78) | 8.27 (7.73-8.81) | |
| No visit | 1.53 (1.48-1.59) | 2.76 (2.69-2.83) | 3.89 (3.81-3.98) | 4.91 (4.82-5.01) | 5.87 (5.76-5.97) | 6.75 (6.64-6.86) | |
| First visit of CMHC | | | | | | | |
| Within 6 weeks | 1.52 (1.47-1.58) | 2.73 (2.66-2.8) | 3.86 (3.77-3.94) | 4.88 (4.78-4.97) | 5.83 (5.73-5.93) | 6.7 (6.59-6.8) | <.001 |
| 6-12 weeks | 2.12 (1.84-2.4) | 3.72 (3.35-4.09) | 5.17 (4.74-5.6) | 6.32 (5.85-6.79) | 7.29 (6.79-7.79) | 8.3 (7.76-8.83) | |
| 12 weeks to 6 months | 2.77 (2.24-3.29) | 4.41 (3.75-5.07) | 5.89 (5.13-6.64) | 7.18 (6.34-8) | 8.39 (7.49-9.27) | 9.41 (8.47-10.34) | |
| No visit | 3.52 (0.44-6.51) | 5.63 (1.76-9.35) | 5.63 (1.76-9.35) | 7.04 (2.74-11.16) | 7.75 (3.24-12.04) | 7.75 (3.24-12.04) | |

Abbreviations: CCI, Charlson Comorbidity Index; N0935, acromioplasty; N0936, acromioplasty and repair of ruptured shoulder rotator cuff, primary repair; N0937, acromioplasty and repair of ruptured shoulder rotator cuff, with myoplasty and tendoplasty; N0938, acromioplasty and repair of ruptured shoulder rotator cuff, complex; M75, shoulder lesions; S46, injury of muscle, fascia and tendon at shoulder and upper arm level; KMHC, Korean medicine healthcare; CMHC, conventional medicine healthcare.

Cumulative incidence rates were estimated using Kaplan-Meier survival analysis, and differences among groups were evaluated using log-rank tests. Results were reported as semiannual cumulative incidence rates of rotator cuff reoperation.

## Risk factors associated with reoperation after RCR

In the unadjusted analysis, significant associations were observed between rotator cuff reoperation and several variables, including sex, age, health insurance type, healthcare institution type, comorbidities, primary RCR type, arthroscopy use, primary diagnosis of primary RCR, receipt of physical therapy, time to the first outpatient visit for KMHC, and time to the first outpatient visit for CMHC. Women (unadjusted HR, 0.92; 95% CI, 0.9–0.95) had a lower risk of rotator cuff reoperation than men. The risk of rotator cuff reoperation was higher in patients aged 45–54 (1.39; 1.31–1.46) or 55–64 (1.46; 1.38–1.54) compared to those aged 19–44. Health insurance type of medical aid/others (1.35; 1.24–1.46), arthroscopy (1.41; 1.11–1.79), and receipt of physical therapy (1.05; 1.02–1.09) were associated with a higher risk of reoperation. Patients who underwent the primary RCR at general hospital (1.15; 1.08–1.23), hospital (1.08; 1.02–1.15), or clinic (1.19; 1.1–1.3) had a higher risk of rotator cuff reoperation compared to those who underwent the primary RCR at tertiary hospital. Comorbidities, as measured by CCI scores of 1 (1.12; 1.08–1.17), 2 (1.24; 1.18–1.3), and 3 or more (1.19; 1.13–1.26), were associated with an increased risk of rotator cuff reoperation. The primary RCR type of N0935 (0.84; 0.81–0.87) was associated with a lower risk of reoperation, while the primary RCR type of N0937 (1.22; 1.16–1.27) and N0938 (1.37; 1.27–1.47) were risk factors increasing rotator cuff reoperation. The risk of rotator cuff reoperation was higher in patients diagnosed with M75 (1.25; 1.17–1.33) or S46 (1.28; 1.19–1.37) than those diagnosed with other conditions at the time of the primary RCR. Time to first outpatient visit for KMHC between 12 weeks and 6 months (1.23; 1.09–1.4) and for CMHC

**Table 3. Factors associated with reoperation after rotator cuff repair.**

| Variables | Unadjusted analysis | | Adjutsed analysis | |
|---|---|---|---|---|
| | uHR (95% CI) | P value | aHR (95% CI) | P value |
| Sex | | | | |
| Men | 1 [Reference] | | 1 [Reference] | |
| Women | 0.92 (0.9-0.95) | <.001 | 0.92 (0.89-0.95) | <.001 |
| Age | | | | |
| 19-44 | 1 [Reference] | | 1 [Reference] | |
| 45-54 | 1.39 (1.31-1.46) | <.001 | 1.31 (1.24-1.39) | <.001 |
| 55-64 | 1.46 (1.38-1.54) | <.001 | 1.27 (1.2-1.35) | <.001 |
| 65 or older | 1.03 (0.96-1.09) | .427 | 0.84 (0.78-0.9) | <.001 |
| Health insurance type | | | | |
| Employee/local | 1 [Reference] | | 1 [Reference] | |
| Medical aid/others | 1.35 (1.24-1.46) | <.001 | 1.32 (1.21-1.43) | <.001 |
| Healthcare institution type | | | | |
| Tertiary hospital | 1 [Reference] | | 1 [Reference] | |
| General hospital | 1.15 (1.08-1.23) | <.001 | 1.12 (1.05-1.2) | .001 |
| Hospital | 1.08 (1.02-1.15) | .009 | 1.12 (1.05-1.19) | <.001 |
| Clinic | 1.19 (1.1-1.3) | <.001 | 1.25 (1.15-1.37) | <.001 |
| CCI score | | | | |
| 0 | 1 [Reference] | | 1 [Reference] | |
| 1 | 1.12 (1.08-1.17) | <.001 | 1.12 (1.07-1.16) | <.001 |
| 2 | 1.24 (1.18-1.3) | <.001 | 1.24 (1.18-1.3) | <.001 |
| 3 or more | 1.19 (1.13-1.26) | <.001 | 1.2 (1.14-1.27) | <.001 |
| N0935 | | | | |
| No | 1 [Reference] | | 1 [Reference] | |
| Yes | 0.84 (0.81-0.87) | <.001 | 0.87 (0.84-0.91) | <.001 |
| N0936 | | | | |
| No | 1 [Reference] | | | |
| Yes | 0.99 (0.96-1.02) | .538 | | |
| N0937 | | | | |
| No | 1 [Reference] | | 1 [Reference] | |
| Yes | 1.22 (1.16-1.27) | <.001 | 1.21 (1.15-1.26) | <.001 |
| N0938 | | | | |
| No | 1 [Reference] | | 1 [Reference] | |
| Yes | 1.37 (1.27-1.47) | <.001 | 1.38 (1.29-1.49) | <.001 |
| Arthroscopy | | | | |
| No | 1 [Reference] | | 1 [Reference] | |
| Yes | 1.41 (1.11-1.79) | .005 | 1.34 (1.05-1.7) | .018 |
| Primary diagnosis | | | | |
| Others | 1 [Reference] | | 1 [Reference] | |
| M75 | 1.25 (1.17-1.33) | <.001 | 1.19 (1.11-1.27) | <.001 |
| S46 | 1.28 (1.19-1.37) | <.001 | 1.16 (1.08-1.25) | <.001 |
| Acupuncture | | | | |
| 5 or less | 1 [Reference] | | | |
| 6 or more | 1.02 (0.93-1.11) | .730 | | |
| Electro-acupuncture | | | | |

*(Continued)*

**Table 3.** (Continued)

| Variables | Unadjusted analysis | | Adjutsed analysis | |
|---|---|---|---|---|
| | uHR (95% CI) | *P value* | aHR (95% CI) | *P value* |
| 5 or less | 1 [Reference] | | | |
| 6 or more | 1.14 (0.97-1.34) | .104 | | |
| Herbal medicine | | | | |
| No | 1 [Reference] | | | |
| Yes | 0.9 (0.8-1.02) | .107 | | |
| Rehabilitation therapy | | | | |
| 5 or less | 1 [Reference] | | | |
| 6 or more | 1.03 (0.99-1.06) | .154 | | |
| Physical therapy | | | | |
| 5 or less | 1 [Reference] | | 1 [Reference] | |
| 6 or more | 1.05 (1.02-1.09) | .002 | 1.03 (1-1.07) | .066 |
| First visit of KMHC | | | | |
| Within 6 weeks | 1 [Reference] | | 1 [Reference] | |
| 6-12 weeks | 1 (0.87-1.14) | .950 | 0.97 (0.85-1.11) | .674 |
| 12 weeks to 6 months | 1.23 (1.09-1.4) | .001 | 1.2 (1.06-1.36) | .003 |
| No visit | 1 (0.9-1.11) | .950 | 0.97 (0.88-1.08) | .628 |
| First visit of CMHC | | | | |
| Within 6 weeks | 1 [Reference] | | 1 [Reference] | |
| 6-12 weeks | 1.25 (1.17-1.34) | <.001 | 1.24 (1.15-1.33) | <.001 |
| 12 weeks to 6 months | 1.43 (1.29-1.59) | <.001 | 1.45 (1.3-1.61) | <.001 |
| No visit | 1.18 (0.65-2.13) | .586 | 1.09 (0.6-1.96) | .786 |
| Mean GVIF | | | 1.067 | |

Abbreviations: HR, hazard ratio; uHR, unadjusted HR; aHR, adjusted HR; CI, confidence interval; CCI, Charlson Comorbidity Index; N0935, acromio-plasty; N0936, acromioplasty and repair of ruptured shoulder rotator cuff, primary repair; N0937, acromioplasty and repair of ruptured shoulder rotator cuff, with myoplasty and tendoplasty; N0938, acromioplasty and repair of ruptured shoulder rotator cuff, complex; M75, shoulder lesions; S46, injury of muscle, fascia and tendon at shoulder and upper arm level; KMHC, Korean medicine healthcare; CMHC, conventional medicine healthcare; GVIF, generalized variance inflation factor.

Statistical significance was evaluated using univariable Cox proportional hazards regression model for unadjusted analysis and multivariable Cox proportional hazards regression model for adjusted analysis. Results were presented as hazard ratios with corresponding 95% confidence intervals.

between 6 and 12 weeks (1.25; 1.17–1.34) or between 12 weeks and 6 months (1.43; 1.29–1.59) after the primary RCR was associated with a higher risk of rotator cuff reoperation compared to first outpatient visit to within 6 weeks.

In the adjusted analysis, risk factors associated with rotator cuff reoperation were similar to those identified in the unadjusted analysis. However, the association between physical therapy and rotator cuff reoperation was not statistically significant after adjusting for confounding variables. Female sex (adjusted HR, 0.92; 95% CI, 0.89–0.95) and primary RCR type of N0935 (0.87; 0.84–0.91) were identified as factors decreasing the risk of rotator cuff reoperation, whereas primary RCR type of N0937 (1.21; 1.15–1.26) and N0938 (1.38; 1.29–1.49) were associated with an increased risk of rotator cuff reoperation. Compared to patients aged 19–44 years, patients aged 45–54 years (1.31; 1.24–1.39) and 55–64 years (1.27; 1.2–1.35) exhibited a higher risk of rotator cuff reoperation, whereas those aged 65 years or older (0.84; 0.78–0.9) had a lower risk of rotator cuff reoperation. Health insurance type of medical aid/others (1.32; 1.21–1.43) and arthroscopy (1.34; 1.05–1.7) were associated with increased risk factors of rotator cuff reoperation. The risk of rotator cuff reoperation was higher in patients who underwent the primary RCR at general hospitals (1.12; 1.05–1.2), hospitals (1.12; 1.05–1.19), or clinics (1.25; 1.15–1.37) compared to those who underwent the primary RCR at tertiary hospitals. Patients

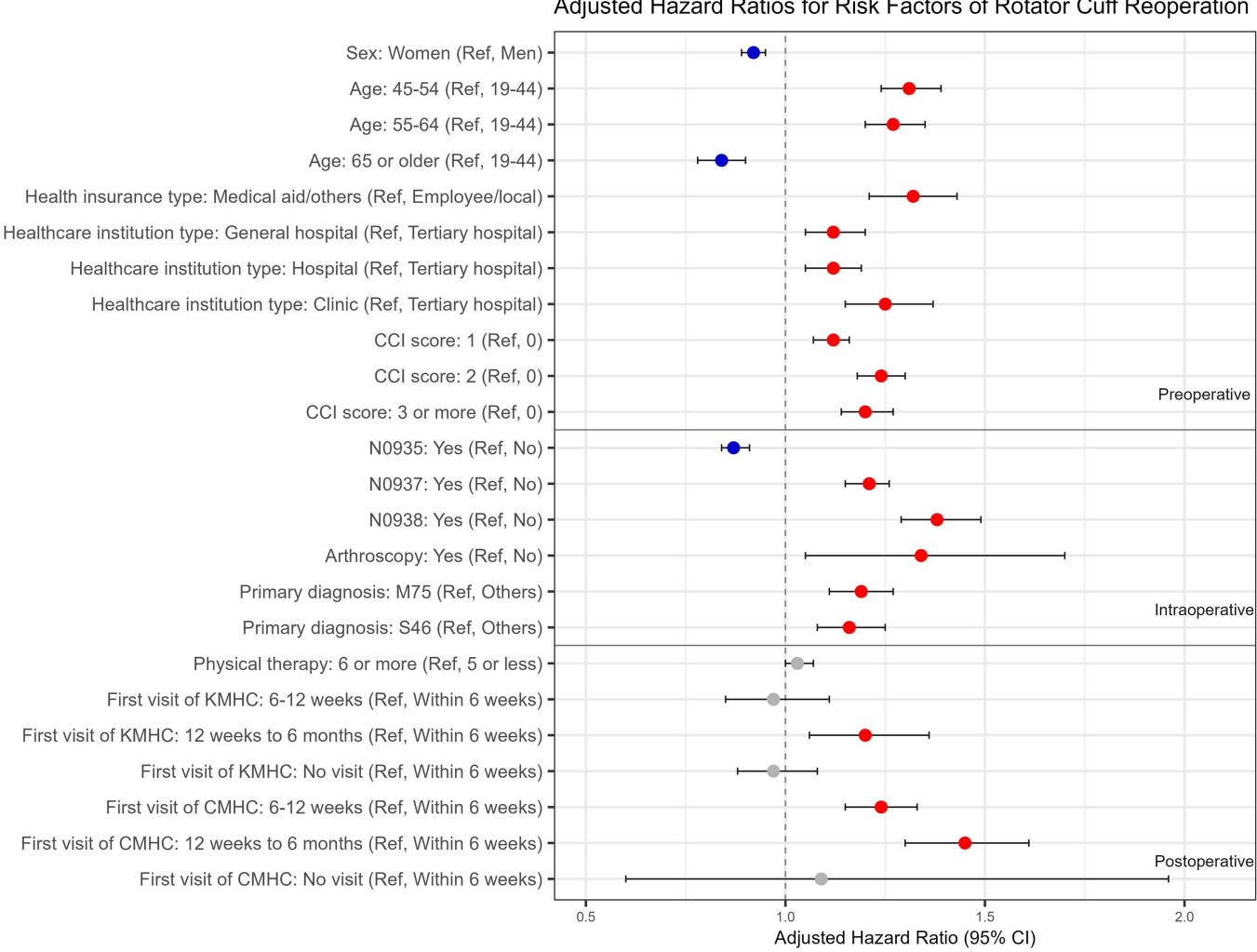

**Fig 2. Hazard ratios with 95% confidence intervals for risk factors of rotator cuff reoperation based on adjusted analysis.**

with comorbidities, as indicated by CCI score of 1 (1.12; 1.07–1.16), 2 (1.24; 1.18–1.3), and 3 or more (1.2; 1.14–1.27), had a higher risk of rotator cuff reoperation than those without comorbidities. Primary diagnoses of M75 (1.19; 1.11–1.27) or S46 (1.16; 1.08–1.25) showed a higher risk of rotator cuff reoperation than other diagnoses at the time of the primary RCR. A higher risk for rotator cuff reoperation was observed in patients who made their first outpatient visit for KMHC between 12 weeks and 6 months (1.2; 1.06–1.36) and for CMHC between 6 and 12 weeks (1.24; 1.15–1.33) or between 12 weeks and 6 months (1.45; 1.3–1.61) after the primary RCR compared to those with first outpatient visits within 6 weeks. Although a few variables showed borderline deviations in the Schoenfeld residual tests, the global test indicated no violation of the proportional hazards assumption. The log–log plots also demonstrated approximately parallel curves across groups, supporting the adequacy of the proportional hazard assumption. The mean GVIF value of the adjusted model was 1.067, confirming the absence of multicollinearity among the predictor variables (Table 3, Fig 2). The supplementary sensitivity analysis using a dichotomized CCI (0 vs. ≥ 1) yielded hazard ratio estimates that were consistent with those of the main model, indicating that the overall findings were robust (Supplementary Table 2).

## Discussion

With the increasing volume of rotator cuff repair (RCR) procedures, identifying factors associated with reoperation has become critical for clinical decision-making. Although several nationwide cohort studies have examined preoperative and intraoperative predictors, their findings have been inconsistent [32–35], and postoperative management factors have been relatively underexamined in relation to reoperation despite growing evidence that they may be associated with clinical recovery and functional improvement [36–39]. Because the need for reoperation reflects a multifactorial process involving patient-related characteristics, surgical complexity, and postoperative care, a comprehensive evaluation across all perioperative stages is necessary. In this context, we conducted a nationwide population-based cohort study using claims data to assess pre-, intra-, and postoperative factors, incorporating real-world conventional and Korean medicine treatments. To our knowledge, this is the first study to investigate reoperation after RCR within a unified analytic framework that integrates all major perioperative domains.

Using the Korean National Health Insurance Database from 2011 to 2021, we identified a cumulative reoperation incidence of 6.82% during the follow-up period from 6 months to 3.5 years after primary repair, reflecting mid- to long-term reoperation rather than overall surgical failure. This rate is lower than previously reported ranges of 9–36% [17,40,41], a difference that is likely attributable to our exclusion of early reoperations within the first 6 months. The cumulative incidence increased most prominently during the first postoperative year and was subsequently examined across individual factor domains to identify factors associated with reoperation.

In this study, preoperative factors associated with reoperation included patient sex, age, health insurance type, healthcare institution type, and comorbidity burden. Male sex showed a higher likelihood of reoperation, consistent with prior studies [32,42,43]. Reoperation risk increased in the 45–64-year age group but declined in those aged ≥65 years, a pattern that may reflect clinical practice wherein older patients with suspected structural failure more often receive conservative treatment or arthroplasty rather than revision surgery [44]. Patients covered by medical-aid insurance showed higher reoperation rates, suggesting a socioeconomic gradient in postoperative outcomes [33,45]. Reoperation risk also increased progressively from tertiary hospitals to clinics, indicating that institutional resources and specialization may be associated with differences in postoperative outcomes. Comorbidity burden demonstrated a clear dose–response relationship, with higher Charlson Comorbidity Index scores associated with increased reoperation risk [32,46,47]. Collectively, these findings indicate that demographic, socioeconomic, and clinical characteristics may play a substantial role in determining postoperative outcomes after RCR.

With respect to intraoperative factors, both the type of RCR procedure and the primary diagnosis at the time of surgery were associated with reoperation. Procedure codes N0937 and N0938—which correspond to repairs involving larger tears, multiple tendons, or complex or revision cases—demonstrated markedly higher reoperation rates. In contrast, N0935, representing isolated acromioplasty, was associated with a lower reoperation rate [25]. However, this finding should be interpreted with caution, as N0935 may be preferentially performed in patients with less severe disease, leading to potential confounding by indication. Therefore, hazard ratios associated with procedural codes should not be interpreted as causal effects. These findings are consistent with prior evidence that initial tear size and structural complexity are major determinants of adverse structural outcomes reported in clinical studies, although such factors were not directly measured in the present claims-based analysis [16,17].

Primary diagnostic codes also showed significant associations. M75 (degenerative shoulder lesions) and S46 (traumatic rotator cuff injuries), which together accounted for the majority of diagnoses at index surgery, were linked to higher reoperation rates compared with other shoulder-related codes. As these codes typically reflect underlying disease severity rather than a direct causal relationship, their association with reoperation should be interpreted with caution.

Lastly, postoperative management options provided within 6 months after surgery were examined to evaluate their association with reoperation. All postoperative treatments identified in the claims data—including rehabilitation, physical therapy, injections, and Korean medicine interventions—were analyzed based on whether each intervention was

administered more than six times. None of the modalities showed a significant association with reoperation after adjustment. Physical therapy initially appeared to be associated with higher reoperation rates in unadjusted analyses, but this association did not remain significant after adjustment.

These findings should also be interpreted in light of several limitations of claims data. Important clinical information—such as postoperative pain, functional status, and imaging findings—was unavailable and may have influenced both treatment utilization and reoperation risk. Patients with persistent symptoms are inherently more likely to seek additional care, which can create apparent associations in unadjusted analyses that do not represent causal effects. Furthermore, because only insurance-covered treatments were captured, the impact of non–insurance-covered postoperative therapies could not be assessed. These factors may explain why postoperative treatment modalities themselves showed weaker associations with reoperation compared with patient characteristics or unmeasured structural factors.

In contrast, the timing of postoperative care demonstrated a clearer association with outcomes. Patients whose first postoperative visit occurred between 12 weeks and 6 months after surgery had a significantly higher reoperation rate compared with those who initiated follow-up within 6 weeks. This pattern was consistent across both conventional and Korean medicine institutions, suggesting that early engagement in postoperative care, rather than the specific modality administered, may be associated with a lower risk of reoperation. We investigated the factors that most affected reoperation after primary repair of the rotator cuff. The major factors relevant to reoperation for rotator cuff tears were patient sex, age, health insurance type, health institutional type, presence of comorbidities, procedure code, and early hospital visits for management ($p < 0.001$; Table 3). Our findings will help in the decision-making process before or after surgery based on patient characteristics, or in counselling patients when to start rehabilitation or other treatment options.

Similar to other nationwide big data studies, our study had some limitations. First, clinical information about pain levels, neurological conditions, quality of life, functional outcomes, radiographic findings, complexity of the operation, and reasons for reoperation were not available at the same time. Second, the HIRA database does not provide laterality information, so we could not fully determine whether follow-up procedures occurred on the same shoulder; therefore, some contralateral surgeries may have been counted as reoperations and should be taken into consideration when interpreting the results. Third, the use of reoperation as the outcome measure may not accurately reflect the true failure or re-tear rate, because not all structural failures lead to reoperation, whereas some reoperations occur for reasons other than re-tears. Because procedural codes cannot distinguish true re-tears from other postoperative indications, we defined the outcome as reoperation to capture all clinically relevant repeat surgeries after the index repair. Fourth, because the follow-up period was defined from 6 months to 3.5 years after surgery, early reoperations within the initial 6 months were not captured. We acknowledge that this may lead to an underestimation of the overall reoperation incidence and may introduce a degree of selection bias. However, early reoperations are generally related to technical issues or acute surgical complications, while our study aimed to evaluate pre-, intra-, and postoperative factors—including postoperative management provided during the initial 6 months—in relation to mid- to long-term outcomes. Therefore, excluding early events was necessary to align the outcome definition with the objective of our analysis. Fifth, because our study used the Korean National Health Insurance Database, only insurance-covered interventions were analyzed. Sixth, although a few variables showed borderline deviations in the proportional hazards assumption test, these deviations were minor, and the global test and log–log plots indicated that the overall proportional hazards assumption was reasonably met. Seventh, although we adjusted for covariates such as age, comorbidities, insurance type, and hospital type, we could not account for several important clinical and surgical factors that are not available in claims data, including tear size and pattern, tendon quality and fatty degeneration, detailed surgical techniques, surgeon experience, and operative time. Previous clinical and imaging studies have shown that tear morphology, tendon quality, surgical technique, and surgeon-related factors are associated with healing and reoperation risk after rotator cuff repair [11,16]. Therefore, residual confounding by these unmeasured factors may remain. Thus, our study should be interpreted with these considerations in mind with these considerations, and further studies investigating each factor separately are needed to determine detailed relationships.

Despite these limitations, our study has notable strengths, using nationwide, population-based claims data to evaluate the incidence and factors associated with rotator cuff reoperation in the general population. The uniform Korean healthcare system enabled consistent identification of surgical procedures and postoperative care patterns, reducing institutional variability. By examining preoperative, intraoperative, and postoperative factors—including treatment timing and the use of Korean medicine—within a single analytic framework, we provided a comprehensive assessment of determinants associated with reoperation.

Through this integrated approach, we identified several key factors associated with reoperation: patient sex, age, health insurance type, health institution type, presence of comorbidities, and procedure code, while early hospital visits for management were associated with a lower risk of reoperation. These findings offer clinically meaningful insights that may support individualized perioperative planning and contribute to optimizing postoperative management strategies for patients undergoing rotator cuff repair.

## Supporting information

**Supplementary Table 1. International Classification of Diseases, 10th revision of shoulder-related disease.**
(DOCX)

**Supplementary Table 2. Sensitivity analysis in adjusted model.**
(DOCX)

## Author contributions

**Conceptualization:** Yee Ran Lyu, Eunkyoung Ahn, Mi Hong Yim.

**Formal analysis:** Mi Hong Yim.

**Funding acquisition:** Changsop Yang.

**Methodology:** Yee Ran Lyu, Mi Hong Yim.

**Supervision:** Changsop Yang.

**Writing – original draft:** Yee Ran Lyu, Mi Hong Yim.

**Writing – review & editing:** Eunkyoung Ahn, Doori Kim, Changsop Yang.

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
