## [Decision Letter · Decision Letter 0]

18 Nov 2025

PONE-D-25-45299Incidence and prognostic factors of reoperation after rotator cuff repair in Korea: A nationwide cohort studyPLOS ONE

Dear Dr. Lyu,

Thank you for submitting your manuscript to PLOS ONE. After careful consideration, we feel that it has merit but does not fully meet PLOS ONE’s publication criteria as it currently stands. Therefore, we invite you to submit a revised version of the manuscript that addresses the points raised during the review process.

We look forward to receiving your revised manuscript.

Kind regards,

Emil George Haritinian, M.D, Ph.D.

Academic Editor

PLOS ONE

Journal Requirements:

4. We note that your Data Availability Statement is currently as follows: All relevant data are within the manuscript and in Supporting Information files.

Additional Editor Comments:

Congratulations on producing a well-written paper that contributes to the existing understanding of reoperation following rotator cuff repairs, with particular relevance to the Korean population.

Respond thoroughly to each reviewer’s query, focusing on these key points:

The acromioplasty procedure code used to search for revisions might overestimate the revision rateExtend the observation period to 0-6 months (early revisions)Re-tear and revision necessity confounding factors should be further discussedClarify funding

Reviewers' comments:

Reviewer's Responses to Questions

**Comments to the Author**

1. Is the manuscript technically sound, and do the data support the conclusions?

Reviewer #1: Yes

Reviewer #2: Partly

2. Has the statistical analysis been performed appropriately and rigorously? 

Reviewer #1: Yes

Reviewer #2: Yes

3. Have the authors made all data underlying the findings in their manuscript fully available?

Reviewer #1: Yes

Reviewer #2: Yes

4. Is the manuscript presented in an intelligible fashion and written in standard English?

Reviewer #1: Yes

Reviewer #2: Yes

5. Review Comments to the Author

Reviewer #1: 1. The author uses surgical codes N0935–N0938 to define whether revision surgery was performed. Among them, N0935 represents simple acromioplasty, which can be done independently without tendon re-tear. Whether it can be equated with repair surgery remains to be verified. This may introduce many cases that do not fall under "repair failure", such as when the tendon heals well after surgery but there is still subacromial impingement pain, and simple acromioplasty is chosen to relieve pressure and pain. This may lead to an overestimation of the "reoperation rate" and increase confounding factors. It is suggested that in addition to the current analysis, a sensitivity analysis using only patients with codes N0936–N0938 be added as supplementary material.

2. Does the HIRA data include information on the affected side? Is it possible that surgeries on the contralateral shoulder of the same patient during the follow-up period were counted as revisions? This should be explained in the limitations section.

3. The study limited the observation period to 6 months to 3.5 years after surgery, excluding reoperations within 6 months. The author acknowledges in the discussion that this may underestimate the incidence rate. However, the fundamental problem with this approach is that early failures are systematically excluded, making the sample selection more "healthy" and potentially leading to selection bias. It is suggested to include a sensitivity analysis covering the 0–3.5-year follow-up period, or at least provide the event rate for 0–6 months to allow readers to judge the extent of underestimation.

4. The information page and the main text have inconsistent statements regarding "Funding". The information page states "No specific funding", while the main text lists KSN2121211. Please unify them.

Reviewer #2: This nationwide cohort study is valuable in quantifying the incidence and prognostic factors of reoperation following RCR in Korea. The topic is clinically relevant, the methodology sound, and the manuscript well written. The finding that delayed initiation of postoperative management (>12 weeks) is associated with a higher reoperation risk is particularly noteworthy and clinically applicable.

To improve the paper’s rigor and transparency, I suggest the following revisions:

Clarify the definition and validation of “reoperation” and discuss possible underestimation from excluding early failures (<6 months).

Expand discussion on unmeasured confounders (tear morphology, surgical technique, surgeon factors).

Specify variables included in the multivariate model and confirm assumption testing.

Enhance figures/tables (confidence bands on survival curves, consistent CI formatting).

Streamline the Discussion to focus on the main findings and reduce redundancy.

With these revisions, the manuscript would make a strong contribution to population-based research on shoulder surgery outcomes.

6. PLOS authors have the option to publish the peer review history of their article (what does this mean?). If published, this will include your full peer review and any attached files.

Reviewer #1: No

Reviewer #2: No

---

## [Author Response · Author response to Decision Letter 1]

7 Dec 2025

Response to Reviewer’s comments

Reviewer #1:

1. The author uses surgical codes N0935–N0938 to define whether revision surgery was performed. Among them, N0935 represents simple acromioplasty, which can be done independently without tendon re-tear. Whether it can be equated with repair surgery remains to be verified. This may introduce many cases that do not fall under "repair failure", such as when the tendon heals well after surgery but there is still subacromial impingement pain, and simple acromioplasty is chosen to relieve pressure and pain. This may lead to an overestimation of the "reoperation rate" and increase confounding factors. It is suggested that in addition to the current analysis, a sensitivity analysis using only patients with codes N0936–N0938 be added as supplementary material.

Response:

Thank you for this valuable and thoughtful comment. We fully agree with the reviewer that simple acromioplasty (N0935) does not necessarily reflect structural failure of the repaired tendon and may therefore contribute to a modest overestimation of the reoperation rate. We also agree that a sensitivity analysis restricted to N0936–N0938 would further clarify the robustness of our findings.

In our study, however, the outcome was intentionally defined as reoperation, rather than structural re-tear, because claims data do not allow differentiation between tendon failure and other postoperative indications. Procedural codes do not capture imaging findings, intraoperative assessments, or surgical intent; thus, reoperations performed for persistent pain, impingement, or other clinical causes cannot be distinguished from those performed for true re-tear. For this reason, we used N0935–N0938 to represent the full spectrum of repeat operative management after the index RCR. To address this concern clearly, we have strengthened the operational definition of reoperation in the Methods section by explicitly describing reoperation as any subsequent surgical intervention following the index repair and by clarifying the use and validation of these codes. To address this issue more explicitly in the manuscript, we added the following clarifying sentence to the Discussion: “Because procedural codes cannot distinguish true re-tears from other postoperative indications, the outcome was intentionally defined as reoperation to capture all clinically relevant repeat surgeries after the index repair.” (Line 404-406) We have also noted the possibility that the inclusion of N0935 may slightly overestimate the absolute reoperation rate.

In addition, because the reviewer’s observation suggested that some expressions in the original manuscript could be interpreted as referring to structural re-tear, we carefully reviewed and revised the wording throughout the manuscript. All outcome-related terminology has now been standardized to explicitly state “reoperation”, ensuring that the intended scope of the study is consistently communicated and that any potential confusion between reoperation and re-tear is avoided.

Although we agree with the reviewer’s suggestion, we are unable to perform additional analyses at this stage. The HIRA research data used in this study were accessible only through a restricted remote-analysis environment during a predefined analysis window. After the approved period ended, the analysis environment was closed under HIRA policy, and no further access or reanalysis is permitted.

To address the reviewer’s concern to the extent possible within these constraints, we have added an alternative supplementary sensitivity analysis that was conducted during the original analysis period. In this analysis, the Charlson Comorbidity Index categories used in the main model (0, 1, 2, and ≥3) were reclassified into a dichotomous form (0 vs. ≥1) to examine the stability of the estimated associations. The resulting hazard ratio estimates remained consistent with those of the primary model, supporting the robustness of our findings. This analysis has been added as Supplementary Table 1, and the corresponding descriptions have been incorporated into the Methods and Results sections as follows:

“In addition to the main multivariate Cox proportional hazards model, we conducted a supplementary sensitivity analysis in which the original four CCI categories (0, 1, 2, and ≥3) were reclassified into a dichotomous variable (0 vs. ≥1) to assess the stability of the estimated associations.” (Lines 211–214)

“The supplementary sensitivity analysis using a dichotomized CCI (0 vs. ≥1) yielded hazard ratio estimates that were consistent with those of the main model, indicating that the overall findings were robust (Supplementary Table 1).” (Lines 315–317)

We sincerely appreciate the reviewer’s insightful suggestion, which has helped us strengthen the clarity of our outcome definition and improve the overall transparency of the manuscript.

2. Does the HIRA data include information on the affected side? Is it possible that surgeries on the contralateral shoulder of the same patient during the follow-up period were counted as revisions? This should be explained in the limitations section.

Response:

Thank you for this important comment. We fully agree that laterality (left/right shoulder information) is an essential factor when interpreting reoperation rates. The HIRA claims database does not provide laterality details for surgical procedures; therefore, we could not determine whether subsequent RCR procedures were performed on the ipsilateral or contralateral shoulder. As the reviewer pointed out, this raises the possibility that contralateral shoulder surgeries may have been counted as revision surgeries.

To address this issue, we have added a clear explanation of this limitation in the revised manuscript. Specifically, we have incorporated the following sentence into the Limitations section:

“Second, the HIRA database does not provide laterality information, so we could not fully determine whether follow-up procedures occurred on the same shoulder; therefore, some contralateral surgeries may have been counted as reoperations and should be taken into consideration when interpreting the results.” (Line 398-401)

We appreciate the reviewer’s insightful suggestion, which has helped improve the clarity and interpretability of our study.

3. The study limited the observation period to 6 months to 3.5 years after surgery, excluding reoperations within 6 months. The author acknowledges in the discussion that this may underestimate the incidence rate. However, the fundamental problem with this approach is that early failures are systematically excluded, making the sample selection more "healthy" and potentially leading to selection bias. It is suggested to include a sensitivity analysis covering the 0–3.5-year follow-up period, or at least provide the event rate for 0–6 months to allow readers to judge the extent of underestimation.

Response:

Thank you for this insightful comment. We agree that excluding reoperations within the first 6 months may lead to an underestimation of the overall reoperation incidence and could introduce a degree of selection bias, as early failures are not captured in our analysis.

However, early reoperations during this period are generally attributable to technical issues or acute surgical complications rather than the patient-, procedure-, or postoperative management factors that our study aimed to evaluate. In addition, reoperations occurring within the first 6 months overlap with the period in which key postoperative management variables are measured. Because postoperative factors are assessed during this same window, including early reoperations as outcomes would create temporal ambiguity and a risk of reverse causation, making it analytically inappropriate to evaluate their association with postoperative factors. Our primary analytic objective was to examine pre-, intra-, and postoperative factors—including postoperative management provided during the initial 6 months—in relation to mid- to long-term outcomes. For this reason, we defined the reoperation outcome from 6 months onward to ensure that early technical failures were not conflated with factors of interest. Therefore, restricting the outcome window to 6–3.5 years was necessary to align the outcome definition with our study purpose.

We fully acknowledge the reviewer’s concern and have clarified this in the revised manuscript. Specifically, we added the following explanation to the Limitations section:

“Fourth, because the follow-up period was defined from 6 months to 3.5 years after surgery, early reoperations within the initial 6 months were not captured. We acknowledge that this may lead to an underestimation of the overall reoperation incidence and may introduce a degree of selection bias. However, early reoperations are generally related to technical issues or acute surgical complications, while our study aimed to evaluate pre-, intra-, and postoperative factors—including postoperative management provided during the initial 6 months—in relation to mid- to long-term outcomes. Therefore, excluding early events was necessary to align the outcome definition with the objective of our analysis.” (Line 406-413)

We appreciate the reviewer’s helpful suggestion, which has allowed us to improve the clarity of our study design and limitations.

Ref) 1. Mandaleson, Avanthi. "Re-tears after rotator cuff repair: current concepts review." Journal of clinical orthopaedics and trauma 19 (2021): 168-174.

2. Felsch, Quinten, et al. "Complications within 6 months after arthroscopic rotator cuff repair: registry-based evaluation according to a core event set and severity grading." Arthroscopy: The Journal of Arthroscopic & Related Surgery 37.1 (2021): 50-58.

4. The information page and the main text have inconsistent statements regarding "Funding". The information page states "No specific funding", while the main text lists KSN2121211. Please unify them.

Response:

Thank you for pointing this out. We apologize for the inconsistency between the information page and the main text regarding the funding source. This study was supported by the Korea Institute of Oriental Medicine (project number KSN2121211), and the correct funding information should appear consistently throughout the manuscript.

We have revised the information page to match the main text and now clearly state the appropriate funding source (KSN2121211). The corrected information has been updated in the revised submission.

We appreciate the reviewer’s careful attention to detail.

Reviewer #2:

1. Clarify the definition and validation of “reoperation” and discuss possible underestimation from excluding early failures (<6 months).

Response:

Thank you for this constructive comment. In our study, “reoperation” was defined as any subsequent surgical intervention performed to manage recurrent or persistent rotator cuff pathology after the index repair. Patients who underwent reoperation were identified as those who received subsequent rotator cuff procedures coded as N0935, N0936, N0937, or N0938 in the HIRA database. These procedural codes are standardized within the Korean National Health Insurance system and have been widely used in nationwide claims-based studies to identify rotator cuff surgeries, supporting their validity for use in administrative data research (Jo et al., 2017). To improve clarity, we have added this operational definition and the supporting reference to the Methods section of the revised manuscript.

“Reoperation was defined as any subsequent surgical intervention to manage recurrent or persistent rotator cuff pathology after the index repair. Patients who underwent reoperation were identified as those who received subsequent rotator cuff procedures coded as N0935, N0936, N0937, or N0938 after the primary RCR. These procedural codes are standardized within the Korean National Health Insurance system and have been used in nationwide claims-based studies to identify rotator cuff surgeries, supporting their validity for use in administrative data research” (Line 148-154)

Regarding the exclusion of early failures, reoperations occurring within the first 6 months were not captured because our follow-up window was defined from 6 months to 3.5 years after surgery. We acknowledge that this may lead to an underestimation of the overall reoperation incidence. Early reoperations are often attributable to technical issues or acute surgical complications, whereas our study aimed to evaluate pre-, intra-, and postoperative factors—including postoperative management during the initial 6 months— in relation to mid- to long-term outcomes. Therefore, excluding early events was necessary to align the outcome definition with the analytic objective of our study.

To clarify this point, we have revised the Limitations section in the manuscript and added the following sentence:

“Fourth, because the follow-up period was defined from 6 months to 3.5 years after surgery, early reoperations within the initial 6 months were not captured. We acknowledge that this may lead to an underestimation of the overall reoperation incidence and may introduce a degree of selection bias. However, early reoperations are generally related to technical issues or acute surgical complications, while our study aimed to evaluate pre-, intra-, and postoperative factors—including postoperative management provided during the initial 6 months—in relation to mid- to long-term outcomes. Therefore, excluding early events was necessary to align the outcome definition with the objective of our analysis.” (Line 406-413)

We appreciate the reviewer’s helpful suggestion, which improved the clarity and transparency of our methodology.

Ref)

1. Mandaleson, Avanthi. "Re-tears after rotator cuff repair: current concepts review." Journal of clinical orthopaedics and trauma 19 (2021): 168-174.

2. Felsch, Quinten, et al. "Complications within 6 months after arthroscopic rotator cuff repair: registry-based evaluation according to a core event set and severity grading." Arthroscopy: The Journal of Arthroscopic & Related Surgery 37.1 (2021): 50-58.

2. Expand discussion on unmeasured confounders (tear morphology, surgical technique, surgeon factors).

Response:

Thank you for this helpful comment. We agree that important unmeasured confounders, such as tear morphology, surgical technique, and surgeon-related factors, may influence retear and reoperation risk after rotator cuff repair. Because our study was based on administrative claims data, we were unable to obtain detailed information on tear size and pattern, tendon quality or fatty degeneration, specific repair configurations, augmentation procedures, surgeon experience, or operative time. Previous clinical and imaging studies have reported that tear morphology, tendon quality, surgical techniques, and surgeon factors are associated with healing, retear, and subsequent surgical outcomes after rotator cuff repair.

To address the reviewer’s suggestion, we have expanded the description of unmeasured confounders in the Limitations section as follows:

“Seventh, although we adjusted for covariates such as age, comorbidities, insurance type, and hospital type, we could not account for several important clinical and surgical factors that are not available in claims data, including tear size and pattern, tendon quality and fatty degeneration, detailed surgical techniques, surgeon experience, and operative time. Previous clinical and imaging studies have shown that tear morphology, tendon quality, surgical technique, and surgeon-related factors are associated with healing, retear, and reoperation risk after rotator cuff repair (11,16). Therefore, residual confounding by these unmeasured factors may remain.” (Line 418-425)

We appreciate the reviewer’s suggestion, which helped us clarify this important limitation more explicitly.

3. Specify variables included in the multivariate model and confirm assumption testing.

Response:

Thank you for this constructive suggestion. To improve clarity, we revised the Methods section to explicitly list the variables included as candidate covariates in the multivariate model.

We also refined our description of the proportional hazards assumption test. In the revi

---

## [Decision Letter · Decision Letter 1]

30 Mar 2026

PONE-D-25-45299R1Incidence and prognostic factors of reoperation after rotator cuff repair in Korea: A nationwide cohort studyPLOS One

Dear Dr. Lyu,

Thank you for submitting your manuscript to PLOS ONE. After careful consideration, we feel that it has merit but does not fully meet PLOS ONE’s publication criteria as it currently stands. Therefore, we invite you to submit a revised version of the manuscript that addresses the points raised during the review process.

Your paper has significantly improved while incorporating the reviewers’ queries. Please resolve the minor issues raised by one of the reviewers. Once these concerns have been handled, I believe the paper will be ready for publication.

We look forward to receiving your revised manuscript.

Kind regards,

Emil George Haritinian, M.D, Ph.D.

Academic Editor

PLOS One

Journal Requirements:

Reviewers' comments:

Reviewer's Responses to Questions

**Comments to the Author**

1. If the authors have adequately addressed your comments raised in a previous round of review and you feel that this manuscript is now acceptable for publication, you may indicate that here to bypass the “Comments to the Author” section, enter your conflict of interest statement in the “Confidential to Editor” section, and submit your "Accept" recommendation.

Reviewer #2: (No Response)

2. Is the manuscript technically sound, and do the data support the conclusions?

Reviewer #2: Yes

3. Has the statistical analysis been performed appropriately and rigorously? 

Reviewer #2: Yes

4. Have the authors made all data underlying the findings in their manuscript fully available?

Reviewer #2: Yes

5. Is the manuscript presented in an intelligible fashion and written in standard English?

Reviewer #2: Yes

6. Review Comments to the Author

Reviewer #2: The authors have adequately addressed the comments raised in the previous round of review. In particular, the revised manuscript provides clearer definitions of the outcome variable, improved transparency regarding the statistical modeling process, and a more explicit discussion of the limitations inherent to claims-based data. These revisions have strengthened the scientific clarity and interpretability of the study.

The study is technically sound and based on a robust analysis of a large, nationwide cohort using an appropriate administrative database. The statistical methods, including survival analysis and multivariable Cox proportional hazards modeling with assumption testing and sensitivity analyses, are suitable for the research objectives and are applied rigorously. The conclusions are appropriately drawn and supported by the data presented, with careful acknowledgment of methodological constraints.

The Data Availability Statement clearly explains the restrictions related to third-party ownership of the data and provides transparent guidance for qualified researchers to access the same dataset through the appropriate institutional channels, in accordance with PLOS ONE data-sharing policies.

The manuscript is well organized and written in clear, standard English. Only minor stylistic polishing may be beneficial, but no substantive language or presentation issues remain that would affect comprehension.

Overall, the study meets the core PLOS ONE publication criteria in terms of originality, methodological rigor, and transparency. However, several methodological clarifications and interpretative refinements are still required to improve scientific clarity and avoid potential misinterpretation of the findings.

1. Definition of “Reoperation” vs. Structural Failure

The authors appropriately clarify that the outcome is reoperation, not structural re-tear, and acknowledge the limitations of claims data. However, despite these clarifications, parts of the Introduction and Discussion still implicitly frame the findings in the context of tendon failure and tear characteristics (e.g., references to tear size and structural complexity). Further tighten the language throughout the manuscript to consistently distinguish clinical reoperation from biological or structural failure. Statements implying causality between postoperative care and tendon integrity should be softened to reflect association rather than mechanism.

2. Inclusion of Acromioplasty (N0935) as Both Index and Outcome Procedure

Although the rationale for including N0935 is explained, this code represents heterogeneous clinical scenarios and may bias both incidence estimates and hazard ratios—particularly given that N0935 appears protective in multivariable analysis. Explicitly caution readers in the Discussion that hazard ratios associated with procedure codes should not be interpreted as causal effects. A short paragraph explaining potential confounding by indication would strengthen interpretability.

3. Exclusion of Early Reoperations (<6 months)

The justification for excluding early reoperations is reasonable from an analytic standpoint. However, this design choice fundamentally alters the interpretation of the reported incidence.Emphasize more clearly (including in the Abstract or Discussion opening) that the reported 6.82% incidence reflects mid- to long-term reoperation, not overall surgical failure. This distinction is crucial for clinical readers.

The manuscript is scientifically sound and suitable for publication in PLOS ONE after minor revisions addressing interpretative clarity and cautious framing of conclusions. The requested changes are primarily editorial and conceptual rather than analytical.

7. PLOS authors have the option to publish the peer review history of their article (what does this mean?). If published, this will include your full peer review and any attached files.

Reviewer #2: No

---

## [Author Response · Author response to Decision Letter 2]

3 Apr 2026

Reviewer Comment 1

Comment:

Definition of “Reoperation” vs. Structural Failure

The authors appropriately clarify that the outcome is reoperation, not structural re-tear, and acknowledge the limitations of claims data. However, despite these clarifications, parts of the Introduction and Discussion still implicitly frame the findings in the context of tendon failure and tear characteristics (e.g., references to tear size and structural complexity). Further tighten the language throughout the manuscript to consistently distinguish clinical reoperation from biological or structural failure. Statements implying causality between postoperative care and tendon integrity should be softened to reflect association rather than mechanism.

Response:

We thank the reviewer for this important and constructive comment. In response, we have carefully revised the manuscript to ensure a consistent distinction between clinical reoperation and structural or biological failure throughout the Introduction and Discussion sections.

1) Clarification of outcome definition (Methods section)

We clarified the definition of reoperation in the Methods section (Lines 152-153) as follows:

Line 152-153: “Reoperation was defined as any subsequent surgical intervention performed after the index RCR, regardless of the underlying clinical indication.”

This revision ensures that the outcome is explicitly defined as a clinical event and not a surrogate for structural failure.

2) Removal of structural failure–oriented framing (Introduction and Discussion)

We revised the Introduction and Discussion sections (Introduction: Lines 58-81; Discussion: Lines 327, 347, 389) to remove or rephrase statements that could implicitly link reoperation to structural failure or tendon re-tear.

Line 58-67: “Despite the outstanding clinical outcomes of rotator cuff repair (RCR) and the evolving surgical techniques, postoperative adverse outcomes remain a concern. In particular, reoperation following RCR has been reported in a wide range of cases (11% to 94%), representing an important clinical event after surgery(8, 9). It should be noted that reoperation does not necessarily indicate structural failure or tendon re-tear, as not all structural failures lead to surgical intervention, and reoperations may occur for various clinical reasons. To date, factors associated with reoperation after RCR remain unclear. Preoperative characteristics such as patient factors, tear and shoulder morphology, intraoperative repair management, and postoperative rehabilitation strategies may be associated with the likelihood of reoperation(10, 11).”

3) Removal of causal language across the manuscript

Throughout the manuscript, expressions implying causality (e.g., “influence,” “lead to,” “reduce”) were systematically replaced with neutral, association-based terms such as “associated with” or “may reflect.”

To further ensure consistency in the interpretation of our findings, we revised the manuscript title from:

“Incidence and prognostic factors of reoperation after rotator cuff repair in Korea: A nationwide cohort study”

to:

“Incidence and factors associated with reoperation after rotator cuff repair in Korea: A nationwide cohort study.”

This change reflects the observational nature of the study and avoids potential causal or predictive implications.

Reviewer Comment 2

Comment:

Inclusion of Acromioplasty (N0935) as Both Index and Outcome Procedure

Although the rationale for including N0935 is explained, this code represents heterogeneous clinical scenarios and may bias both incidence estimates and hazard ratios—particularly given that N0935 appears protective in multivariable analysis. Explicitly caution readers in the Discussion that hazard ratios associated with procedure codes should not be interpreted as causal effects. A short paragraph explaining potential confounding by indication would strengthen interpretability.

Response:

We thank the reviewer for this insightful comment. We agree that the inclusion of N0935 (acromioplasty) may reflect heterogeneous clinical scenarios and that the observed association should be interpreted with caution. In response, we have revised the Discussion section (Lines 361-367) to explicitly address potential confounding by indication. Specifically, we clarified that N0935 may be preferentially performed in patients with less severe disease or smaller tears, which may lead to an apparent protective association with reoperation. We further emphasized that hazard ratios associated with procedural codes should not be interpreted as causal effects, but rather as reflecting underlying differences in patient characteristics or disease severity. In addition, we noted that important clinical variables such as tear size, tendon quality, and surgical complexity are not available in the claims data, and therefore procedural codes may act as proxies for unmeasured severity, contributing to residual confounding.

Line 361-367: However, this finding should be interpreted with caution, as N0935 may be preferentially performed in patients with less severe disease, leading to potential confounding by indication. Therefore, hazard ratios associated with procedural codes should not be interpreted as causal effects. These findings are consistent with prior evidence that initial tear size and structural complexity are major determinants of adverse structural outcomes reported in clinical studies, although such factors were not directly measured in the present claims-based analysis.

Reviewer Comment 3

Comment:

Exclusion of Early Reoperations (<6 months)

The justification for excluding early reoperations is reasonable from an analytic standpoint. However, this design choice fundamentally alters the interpretation of the reported incidence. Emphasize more clearly (including in the Abstract or Discussion opening) that the reported 6.82% incidence reflects mid- to long-term reoperation, not overall surgical failure. This distinction is crucial for clinical readers.

Response:

We thank the reviewer for this important comment. We agree that excluding early reoperations affects the interpretation of the reported incidence. In response, we have revised both the Abstract (Lines 30-31) and the opening of the Discussion (Lines 337-338) to clearly state that the reported 6.82% incidence reflects mid- to long-term reoperation (from 6 months to 3.5 years after surgery), rather than overall surgical failure. We believe this clarification improves the clinical interpretability of our findings and aligns with the reviewer’s recommendation.

Line 30-31, Line 337-338: Using the Korean National Health Insurance Database from 2011 to 2021, we identified a cumulative reoperation incidence of 6.82% during the follow-up period from 6 months to 3.5 years after primary repair, reflecting mid- to long-term reoperation rather than overall surgical failure.

---

## [Decision Letter · Decision Letter 2]

11 May 2026

Incidence and factors associated with reoperation after rotator cuff repair in Korea: A nationwide cohort study

PONE-D-25-45299R2

Dear Dr. Lyu,

We’re pleased to inform you that your manuscript has been judged scientifically suitable for publication and will be formally accepted for publication once it meets all outstanding technical requirements.

Kind regards,

Emil George Haritinian, M.D, Ph.D.

Academic Editor

PLOS One

Reviewers' comments:

Reviewer's Responses to Questions

**Comments to the Author**

1. If the authors have adequately addressed your comments raised in a previous round of review and you feel that this manuscript is now acceptable for publication, you may indicate that here to bypass the “Comments to the Author” section, enter your conflict of interest statement in the “Confidential to Editor” section, and submit your "Accept" recommendation.

Reviewer #2: All comments have been addressed

2. Is the manuscript technically sound, and do the data support the conclusions?

Reviewer #2: Yes

3. Has the statistical analysis been performed appropriately and rigorously? 

Reviewer #2: Yes

4. Have the authors made all data underlying the findings in their manuscript fully available?

Reviewer #2: Yes

5. Is the manuscript presented in an intelligible fashion and written in standard English?

Reviewer #2: Yes

6. Review Comments to the Author

Reviewer #2: The authors have adequately addressed the concerns raised in the previous round of review. The revised manuscript now clearly distinguishes clinical reoperation from structural failure or tendon re-tear, and the interpretation of the reported 6.82% incidence has been appropriately clarified as mid- to long-term reoperation from 6 months to 3.5 years after primary repair, rather than overall surgical failure.

The authors have also appropriately added caution regarding the interpretation of procedure codes, particularly N0935, and have acknowledged the possibility of confounding by indication and residual confounding due to the inherent limitations of claims-based data. These revisions substantially improve the interpretability and clinical relevance of the findings.

Overall, the manuscript is technically sound, the statistical analysis is appropriate, and the conclusions are supported by the data. I have no further major concerns and recommend acceptance.

7. PLOS authors have the option to publish the peer review history of their article (what does this mean?). If published, this will include your full peer review and any attached files.

Reviewer #2: No

---

## [Editor Report · Acceptance letter]

PONE-D-25-45299R2

PLOS One

Dear Dr. Lyu,

I'm pleased to inform you that your manuscript has been deemed suitable for publication in PLOS One. Congratulations! Your manuscript is now being handed over to our production team.

Kind regards,

on behalf of

Dr. Emil George Haritinian

Academic Editor

PLOS One